# Antibiotic Drug Resistance Pattern of Uropathogens in Pediatric Patients in Pakistani Population

**DOI:** 10.3390/antibiotics12020395

**Published:** 2023-02-15

**Authors:** Zakia Iqbal, Ahsan Sattar Sheikh, Anwaar Basheer, Hadiqa tul Hafsa, Mehboob Ahmed, Anjum Nasim Sabri, Samiah Shahid

**Affiliations:** 1Institute of Molecular Biology and Biotechnology, The University of Lahore, Lahore 54000, Pakistan; 2The Children’s Hospital, University of Child Health Sciences, Lahore 54660, Pakistan; 3Department of Microbiology, Shaikh Zayed Hospital, Lahore 54600, Pakistan; 4Institute of Microbiology and Molecular Genetics, Quaid-e-Azam Campus, University of the Punjab, Lahore 54590, Pakistan

**Keywords:** uropathogens, *Escherichia*, *Klebsiella*, antibiotic resistance

## Abstract

The common prevalent diseases in the age of 0 to 6 are related to urinary tract infections. If not properly diagnosed, they will lead to urological and nephrological complications. Uropathogens are developing resistance against most drugs and are harder to treat. A study was done on the inpatients and outpatients of the two hospitals located in Lahore. A total of 39,750 samples that were both male and female were collected. *Escherichia* and *Klebsiella* were found in 234 samples based on biochemical characterization, growth on CLED agar, and white blood cell/pus cell (WBC) microscopy. In comparison to males, female samples had a higher number of uropathogens (1:1.29). From the samples of Shaikh Zayed Hospital (SZH), the ratio of *Klebsiella* to *Escherichia* (1:1.93) was reported, while this ratio was 1.84:1 from the Children Hospital (CH). The incidence of UTI was higher in the month of September. Randomly selected *Escherichia* and *Klebsiella* were verified via a 16S rRNA sequence. Antibiotic resistance profiling of isolated bacterial strains was done against 23 antibiotics. The most efficient antibiotics against *Klebsiella* and *Escherichia* were colistin sulphate (100% sensitivity against bacteria from CH; 99.3% against strains from SZH) and polymyxin B (100% sensitivity against strains from SZH; 98.8% against strains from CH). Sensitivity of the total tested strains against meropenem (74%, SZH; 70% CH), Fosfomycin (68%, SZH; 73% CH strains), amikacin (74% SZH; 55% CH), and nitrofurantoin (71% SZH;67% CH) was found, Amoxicillin, ampicillin, and cefuroxime showed 100 to ≥90% resistance and are the least effective.

## 1. Introduction

Children’s urinary tract infections are among the most prevalent bacterial illnesses, and primary care pediatricians are quite concerned about them. UTIs are either hospital- or community-acquired and have resulted in an increase in visits to the pediatric emergency department, which is also a cause of hospitalization. UTIs became more frequent in infants and young children than in adults. How much UTI is prevalent in pediatrics is determined by the age of the child, gender, location, population load on the community, geographical region, and other community-based factors. Infection rates are higher in females than in male children. In neonates and young children, the symptoms and signs of UTIs are nonspecific, and its diagnosis is complex. A delay in it can result in the different morbidities such as bacteremia, sepsis, renal disorders, kidney damage or failure, hypertension, etc. For the prevention of such morbidities, early diagnosis, adequate treatment, and follow-up are crucial problems [1,2,3,4,5,6,7,8]. Gram-negative rods are more common and the primary cause of urinary tract infections (UTIs) compared to Gram-positive bacteria [9]. The pathogens causing UTIs mostly include *Escherichia*, *Klebsiella*, *Proteus*, *Staphylococcus*, *Streptococcus*, *Enterobacter*, *Pseudomonas,* and *Enterococcus,* and others [10,11,12]. In various studies, it is reported that the main etiopathogenic agent is *E. coli* [10,13,14].

Uropathogens have strong invasion, adhesion, and virulence abilities. These properties, combined with the emergence of increased antimicrobial resistance in them, are becoming a global concern [9]. Many pediatric uropathogens have been shown to display varied levels of a resistance/sensitivity pattern to antibiotics, including aminoglycosidase, penicillins, cephalosporins, carbapenem, trimethoprim/sulfamethoxazole, nalidixic acid, nitrofurantoin, phosphomycin, quinolones, and polymyxins [3,5,15].

The current study was based on the assumption that age, locality, and gender have an impact on antibiotic resistance patterns in uropathogens. In this work, the prevalence of uropathogenic *Escherichia* and *Klebsiella* spp. is correlated with the occurrence and emergence of multiple-drug resistant (MDR), extensively drug resistant (XDR), and pan-drug resistant (PDR) bacteria in hospitals and the community of Pakistan.

## 2. Results

### 2.1. Sample Collection

Two hospitals from Lahore, Pakistan were selected for the sampling. From Shaikh Zayed hospital (SZH), urine samples were collected from June to September 2021, while from the Children Hospital (CH), the sampling time was September to November 2021 and January to February 2022. The study was conducted on the pediatric patients, aged 0–6 years, from different indoor and outdoor patients of the two hospitals (Table 1). Fresh voided midstream urine samples were collected in sterile wide-mouth containers from both male and female patients. The samples were immediately transported to the pathology laboratories of the hospitals for further processing. The total samples collected from two hospitals were 39,750. After a microscopic examination of the urine samples, samples having less than 5 WBC (white blood cells/pus cells) under a magnification of 400X were excluded from the study. The remaining samples (≥5 WBC) were cultured on CLED agar plates. The samples showing no growth on CLED agar were also excluded. Colonies which appeared on CLED agar plates were characterized on the basis of colour and colony. From the CLED agar plates, a total of 234 bacterial strains were identified as *Escherichia* sp. and *Klebsiella*.

Out of these 234 strains, the prevalence of *Escherichia* sp. and *Klebsiella* sp. is 62.40% at SZH and 37.60% at CH. From SZH, 42.46% samples from male and 57.53% samples from female were positive for *Escherichia* sp. and *Klebsiella*. From CH, 45.45% of samples from males and 54.54% of samples from females were positive for *Escherichia* sp. and *Klebsiella*. More than 86.61 percent of the UTI-positive samples from Shaikh Zayed Hospital had a bacterial count of >10^5^ CFU/mL. This percentage in CH was 92.04 percent, while 7.95% and 14.38%, respectively, of the CH and SZH samples exhibited reduced bacterial counts of >10^4^ CFU/mL (Table 1).

From SZH, a total of 26 IPD/OPD sources were screened for the presence of the *Escherichia* sp. and *Klebsiella* as uropathogens, while from CH, a total of 14 IPD/OPD sources were screened. The highest number of positive samples was obtained from patients of SZH, who visited the emergency ward (35 samples), which was followed by the urology IPD (14 samples positive) (Figure 1b). From CH, the highest number was reported from the urology IPD (24) followed by the nephrology IPD (18 samples) (Figure 1a).

When UTI-positive samples from both hospitals were compared with respect to the new cases reported in each month, it was observed that the incidence of *Escherichia* sp. and *Klebsiella* was higher in the month of September. The incidence of *Klebsiella* in each month was higher in the samples from CH as compared to the samples from SZH. The number of *Klebsiella* reported in each case in the samples from CH was higher in females as compared to the males (Figure 2).

### 2.2. Morphological and Biochemical Characterization of Antibiotic-Resistant Escherichia and Klebsiella Isolated from UTIs

A total of 234 (146 from SZH and 88 from CH) bacterial isolates, which were isolated from UTI in pediatrics, were further screened for confirmation of *Escherichia* and *Klebsiella* species by performing their morphological and biochemical tests. Purification of bacterial isolates was done on several types of CLED-inhibited colonies. A total of 234 bacterial strains were designated names according to the hospital. For SZH, the isolates were coded as prefix MAZKS (1–146) and for CH strains, they were assigned names with the prefix MAZKC- (147–234). The results of these isolates are being reported in the current study.

#### 2.2.1. Shaikh Zayed Hospital

The morphological characterization of 146 strains obtained from samples of different IPD and OPD of SZH was performed. Gram-staining was performed, and results revealed that all the strains were Gram-negative. Cells were non-motile in 14 strains. All strains showed negative results for oxidase activity. All these strains were positive for catalase, triple sugar iron test, and gas production. Positive urease activity and citrate utilization were observed in 14 strains. Indole test was also negative in 14 strains. On the basis of the above result, 57 bacterial isolates in males and 75 in females were identified as *Escherichia*; five in males and nine in females were identified were as *Klebsiella* spp. (Figure 3).

#### 2.2.2. Children Hospital

The morphological characterization of 88 strains obtained from samples of different IPD and OPD of CH was performed. All the strains were Gram-negative. Only 38.27% of the strains showed positive results for motility. All strains showed negative results for oxidase activity. All these strains were positive for catalase, triple sugar iron test, and gas production. Urease activity and citrate utilization were negative in 38.27% of the strains. Indole test was positive in 31 strains out of 88. On the basis of the above result, 14 bacterial isolates in males and 17 in females were identified as *Escherichia,* and 26 in males and 31 in females were identified as *Klebsiella* spp. (Figure 3).

### 2.3. Antibiotic Resistance/Susceptibility Profile

All the isolated *Escherichia* & *Klebsiella* from SZH and CH were checked against the different antibiotics. All bacterial strains (*Escherichia* and *Klebsiella*) showed 100% resistance against amoxicillin (SZH), ampicillin (both SZH and CH), and cefuroxime (SZH). Approximately ≥90% of the tested strains showed resistance against amoxicillin (SZH), co-amoxiclav, ceftroxone, cefotaxmine, cefixime (for both SZH and CH), ciprofloxin, and cefuroxime (SZH). In the case of collistine sulphate (SZH) and polymyxin B (CH), 100% susceptibility in the tested strains was recorded. Bacterial isolates from both hospitals displayed different resistance and sensitivity percentages for the other tested antibiotics (Figure 4). In the case of SZH, bacterial strains were also tested for other antibiotics where, against imipenem, 28% of the bacterial strains showed resistance and 72% showed sensitivity. Against gentamycin, 37% of the tested bacterial strains showed resistance. Similarly, approximately 77 (nalidixic acid), 81 (pipedemic acid, norfloxaxin), 55 (tobramycin), 94 (co-trimoxazole), and 61% (amoxicillin/clavulanic acid) of the total strains showed resistance against the respective antibiotics (Figure 4).

#### 2.3.1. Shaikh Zayed Hospital

When antibiotic-resistant bacterial strains from different age groups were analysed, it was found that the ratio of male and female samples is more or less the same. The bacterial strain MAZKS143 which showed resistance to 90% of the tested antibiotics was identified as *Escherichia* sp.; MAZKS3, MAZKS10, MAZKS11, MAZKS31, MAZKS38. MAZKS42, MAZKS57, MAZKS61, MAZKS76, MAZKS78, MAZKS79, MAZKS109, MAZKS114, MAZKS122, MAZKS130, MAZKS140, and MAZKS144 showed resistance against equal to or greater than 80% of the antibiotics tested. Among the above-mentioned strains, only MAZKS122 is reported as *Klebsiella.* For the remaining strains, the percentages vary between resistance, sensitivity, and intermediate level. In the case of MAZKS69, MAZKS95–96, and MAZKS111, the sensitivity to total antibiotics tested was ≥80% (Figure 5).

#### 2.3.2. Children Hospital

All bacterial strains were tested against 23 antibiotics. When antibiotic-resistant bacterial strains were isolated from the samples of children under the age of one year, it was noted that all belong to females, while for the other age groups, bacterial strains were isolated from both male and female samples. Bacterial strains MAZKC 174, MAZKC 153, MAZKC 195, MAZKC 217, and MAZKC 223 resisted 90% of the tested antibiotics and all of them were identified as *Klebsiella*. MAZKC 156, MAZKC 172, MAZKC 175, MAZKC 204–208, MAZKC 213–214, MAZKC 2018–222, MAZKC 224, and MAZKC 229 showed resistance against equal to or greater than 80% of the antibiotics tested and, in these strains, only four belong to *Escherichia* sp. For the remaining strains, the percentages vary between resistance, sensitivity, and intermediate level. In the case of MAZKC197 and MAZKC233–234, the sensitivity to total antibiotics tested was more than 80% (Figure 6).

## 3. Discussion

Urinary tract infections among neonates, infants, and young children are reported as a common disease in hospitalized and external consultation units. It also has the ability to reoccur. Taking this into account, two hospitals from different locations of Lahore were selected. This study is the continuity of the study of Iqbal et al. (2021). Samples from patients between the age of 5 days up to 6 years were collected. On the basis of gender, male and female samples were segregated. It was observed that the prevalence and incidence of UTI is higher in females than males. The female samples were 15.06% more than males in SZH. From CH, the female samples were reported as 9.1% more than the male samples. In a previous study, the male-to-female ratio from Children Hospital was 1:1.04 (2% more males than females) (Iqbal et al.) [16]. It is reported worldwide that UTI is more prevalent in females than in males [2,5,17]. In both hospitalized and outdoor female patients ≥age of five, UTI is reported with a high frequency than in males [1]. The reason might be an anatomical difference in the male and female urethra. The female urethra is straight, smaller in size and diameter, and not differentiated into parts. In females, it open to the outside between the clitoris and vagina and is the only urinary system. The invasion and adhesion to epithelial layers of the uropathogens are easy in females as compared to males. Due to the close proximity of the vagina, anus, and urinal openings in females, urinal infections are prevalent [10,18,19]. The highest number of UTI-positive samples were obtained from patients from the emergency and urology IPD. Kasanga et al. [3] described the distribution of bacterial isolates ward-wise and found that the maximum was from emergency wards, with the least from internal medicine wards. Females are reported more in both groups, i.e., hospitalized group and external consultation group [1].

When UTI-positive samples from both hospitals were compared, it was observed that *Escherichia* was more prevalent from SZH, whereas *Klebsiella* was reported more from the CH. The infection may be community-acquired or may be hospital-based [6]. Urinary tract infections are the most common infection in pediatrics and *the* most prevalent [20] etio-pathogen agent is reported as *E. coli* [1,2,5,8,17,18,21,22,23]. Ganesh et al. [18] reported that the most frequently detected bacteria was *E. coli*, followed by *Klebsiella*. A previous study of Iqbal et al. [16] also showed *Escherichia* sp. as 63% of the total sample, with *Klebsiella* as 37%. It was observed that the incidence of *Escherichia* sp. and *Klebsiella* was higher in the month of September as compared to the other months. It might be due to the fact that in September, the minimum (28 °C) and maximum (36 °C) temperatures were ideal for the growth of bacteria. Iqbal et al. [16] conducted a study between June 2018 to February 2019 and reported that the prevalence of *Escherichia* was 63% and that of *Klebsiella* was 37%. All the isolated *Escherichia* and *Klebsiella* from SZH and CH were checked against the different antibiotics. All biochemically reported *Escherichia* and *Klebsiella* showed 100% resistance against amoxicillin (SZH), ampicillin (both SZH and CH), and cefuroxime (SZH). Approximately ≥90% resistance was found against amoxicillin (SZH), co-amoxiclav, ceftroxone, cefotaxmine cefixime (for both SZH, CH), ciprofloxin, and cefuroxime (SZH). Bacterial isolates from both hospitals showed different levels of resistance and sensitivity, for the tested antibiotics. In the case of SZH, bacterial strains also showed resistance against imipenem and gentamycin, while for the Children Hospital, nalidixic acid, pipedemic acid, norfloxaxin, tobramycin, co-trimoxazole, and amoxicillin/clavulanic acid showed different resistance profiles. In the case of collistine sulphate (SZH) and polymyxin B (CH), 100% susceptibility by the tested strains was recorded. According to a one-year (2018–2019) study conducted by Iqbal et al. [16], more than 90% of the uropathogenic strains examined had antibiotic resistance to cefuroxime, cefixime, and cefotaxime. Additionally, co-amoxiclav, ceftazidime, ceftriaxone, ciprofloxacin, nalidixic acid, norfloxacin, pipemidic acid, and co-trimoxazole were found to be ineffective against more than 80% of the uropathogenic strains. In the present study, resistance against amoxicillin, ampicillinm and cefuroxime is increased over a period of time.

When the data of resistance of a year (2000–2004) was compared with the resistances observed in 2015–2019, a shift in the percentage resistance to antibiotics in *E. coli* from 16% to 36% was reported [24]. *E. coli* strains showed an increase in resistance to amoxicillin/clavulanic acid, with an average increase of 2.0 percent per year, and 1.1% per year for cephalosporins [25]. Over a period of five years, a shift in *E. coli* resistance to nitrofurantoin (10.7%), trimethoprim-sulfamethoxazole, and ampicillin (44.2%) was recorded. Cephalosporins, in contrast, have maintained low resistance levels of antibiotic resistance in this study [5]. A study from General Hospital, Lahore, Pakistan also showed a different pattern of occurrence of antibiotic resistance in different uropathgens. Most of the isolated UTI pathogens are 72–95% resistant to ampicillin, cotrimoxazole, and cephalexin. Against cephalosporins, intermediate resistance is reported, while sensitive behavior towards amikacin, nitrofurantoin, and ciprofloxacin is also reported by this study [20].

Studies during September 2020–December 2020, at the Institute of Kidney Diseases Hospital, Multan, Pakistan, reported that oxazolidinones drugs were more effective against Gram-positive cocci. β-lactams, especially carbapenems and penicillins, were found to effectively kill the Gram-negative rods. They further reported that nitrofurantoin and fosfomycin were most effective against both types of pathogens [26]. Gentamycin inhibits protein synthesis in *E. coli*. Approx. 44% of the *E. coli* isolated as uropathogen bacteria exhibited maximum resistance [27]. Most of the Enterobacteriaceae members from pediatric UTIs were found to be resistant to carbapenem [25]. Epidemiological studies about urinary tract infections and antibiotic resistances by Ganesh et al. [18] reported high drug resistance among isolates, with most of them resistant to ampicillin and co-trimoxazole. The least resistance among isolates was observed against amikacin and nitrofurantoin [18]. Antibiotic-resistant *Escherichia* and *Klebsiella* were recorded in all age groups and even neonates and infants showed a high resistance to the antibiotics tested. It is alarming that children under the age of one week or less than a month also showed resistances to antibiotics. In urinary tract infections, the age, race, gender, status of circumcision, and pattern of antibiotics may affect etiological agent/pathogen emergence [6,28,29]. Until 2 years of age, the prevalence of UTIs in girls and boys is 1:1.09 [29]. The highest baseline prevalence of UTIs was found in females younger than 12 months of age and uncircumcised male newborns younger than 3 months of age. Up to 8% of children at the age of 1 month to 11 years (up to 8%) may experience at least one UTI. Recurring infections were also observed in 30% of infants and young children within 6 months to one year of the initial infection [21,30]. Unsuitable and frequent use of antimicrobial agents, sanitary conditions, living behaviors, localities, and overcrowding may lead to the development of resistance against most antibiotics, which leads to difficulties in treatment conditions. In many cases, antibiotics can be easily purchased from the market without a prescription, which may be one cause of the high resistance rates to antimicrobial agents.

## 4. Materials and Methods

### 4.1. Study Area

The study area was the two hospitals (Table 2).

Both hospitals have well-equipped pathology units with all facilities of microbiology and chemical pathology analysis. These hospitals diagnose many indoor and outdoor patients.

### 4.2. Inclusion Criteria

The study included all pediatric patients with UTI who were between the ages of 0 and 6 years.

### 4.3. Exclusion Criteria

Patients older than six years and without UTI symptoms were not included in this study.

### 4.4. Sample Collection

Two hospitals from Lahore, Pakistan were selected for the sampling. From Shaikh Zayed Hospital (SZH), urine samples were collected during June to September 2021, while from Children Hospital (CH), sampling time was September to November 2021 and January to February 2022. The study was conducted on the pediatric patients, aged 0–6 years, from different indoor and outdoor patients of the two hospitals. Fresh voided midstream urine samples were collected in sterile wide-mouth containers from both male and female patients. The samples were immediately transported to pathology laboratories of the hospitals for further processing. Following Cappuccino and Sherman [31], centrifugation of urine sample at 10,000× *g* (5 min) was performed. After staining, white blood cell counts (pus cells) were observed and counted at 400X. Urine samples having ≥5 white blood cells (WBC) under 400X were assumed positive for UTI. For screening and isolation of *Escherichia* and *Klebsiella* spp., cysteine lactose electrolyte-deficient agar was used. After incubation at 37 °C for 72 h, colonies which appeared as yellow, opaque, or slightly deeper in color were initially recorded as *Escherichia* spp., while large mucoid yellow or yellow-white colonies showed similarity with *Klebsiella* spp.

### 4.5. Morphological and Biochemical Characterization

After subculturing, purified selected colonies were characterized morphologically and biochemically. Gram staining, cell morphology, motility, oxidase, catalase, urease, citrate utilization, indole production, and triple iron sugar tests were performed [31]. *Escherichia* showed negative results for oxidase, urease, and citrate utilization, while showing positive results for catalase, indole, and TSI. Cells of *Escherichia* were motile and Gram-negative. *Klebsiella* showed negative results for oxidase and indole test, while urease, citrate utilization, catalase, and TSI was positive. Cells of *Klebsiella* were non-motile and Gram-negative.

### 4.6. Antibiotic Resistance Test

The disc diffusion method of Kirby–Bauer was employed to evaluate the antibiotic resistance pattern. The commercially available antibiotic paper discs of 4 mm size enlisted in Table 3 were loaded on MH plates. The antibiotic paper discs were purchased from Abtek Biologicals Ltd. Liverpool, UK. Plates were incubated at 37 °C for 24 h. The diameter of bacterial growth inhibition zone around the antibiotic paper disc was observed. The *E. coli* ATCC 23,509 and *E. coli* ATCC 25,922 were used as a control. Standard diameter zone provided by Clinical and Laboratory Standard Institute (CLSI) was followed for categorization of resistant, intermediate, and sensitive [32].

Following worksheets and criteria of Magiorakos et al. [33] for categorizing isolates of Enterobacteriaceae, antibiotic resistant strains were classified as MDR, XDR, and PDR.

## 5. Conclusions

It is clear from the analysis of the antibiotic resistances in pediatric UTIs that *E. coli* and *Klebsiella* from both hospitals displayed high-level antimicrobial resistance. MDR, XDR, and ESBL bacterial strains are alarmingly common in newborns and other age groups of children from 0 to 6 years old. The treatment of these strains is challenging and difficult. The evaluation of uropathogen prevalence and antibiotic susceptibility is crucial for the appropriate management of UTIs in pediatrics.

## Figures and Tables

**Figure 1 antibiotics-12-00395-f001:**
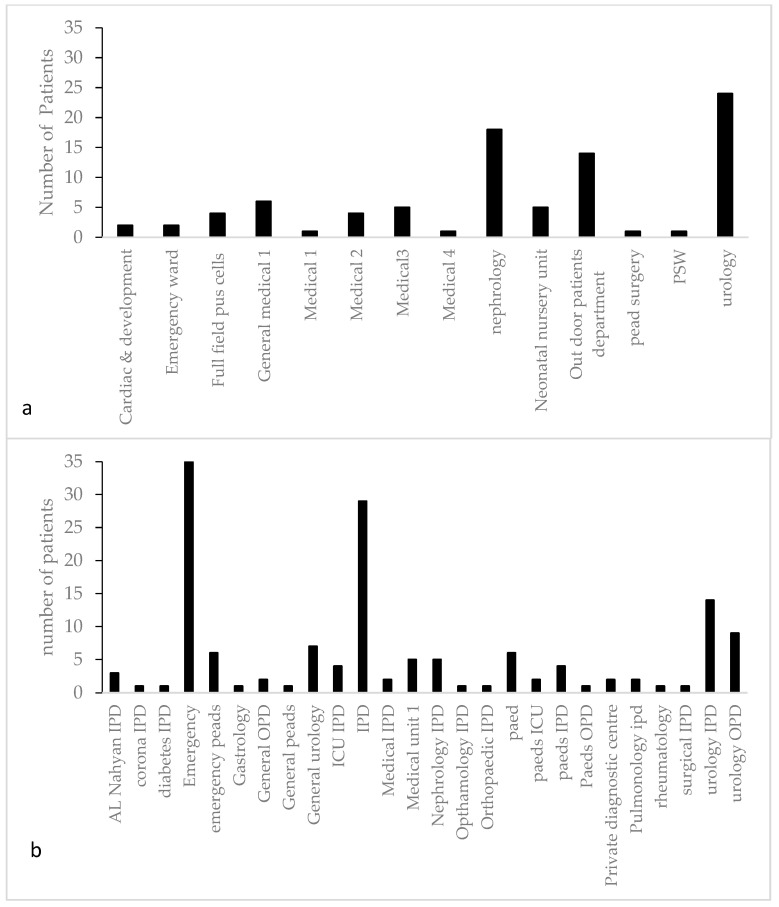
Ward-wise distribution of pediatric patients’ (aged 0–6 years) samples showing positive results for *Escherichia* and *Klebsiella* on CLED agar: (**a**) from Children Hospital; (**b**) from Shaikh Zayed Hospital.

**Figure 2 antibiotics-12-00395-f002:**
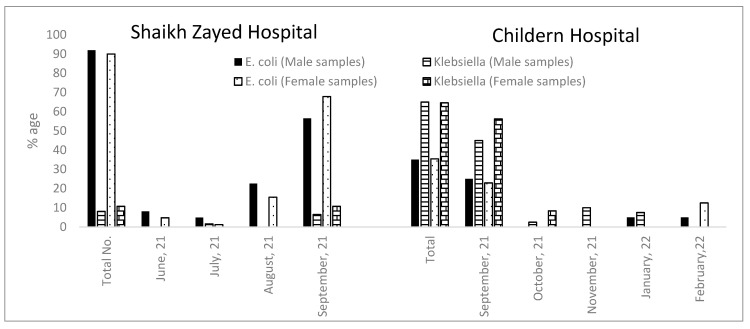
Month-wise distribution of *Escherichia* and *Klebsiella* from samples of pediatric patients (aged 0–6 years) at Shaikh Zayed and Children Hospital, Lahore, Pakistan.

**Figure 3 antibiotics-12-00395-f003:**
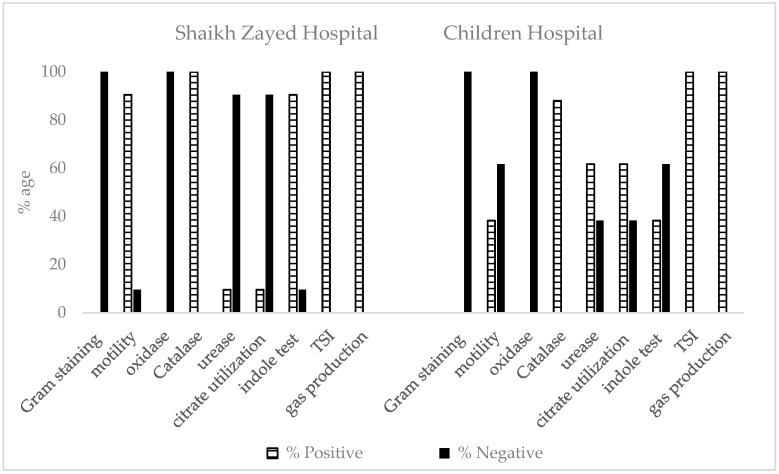
Morphological and biochemical characterization of bacterial strains isolated from pediatric patients (aged 0–6 years) at Shaikh Zayed Hospital, Lahore, suffering from UTI.

**Figure 4 antibiotics-12-00395-f004:**
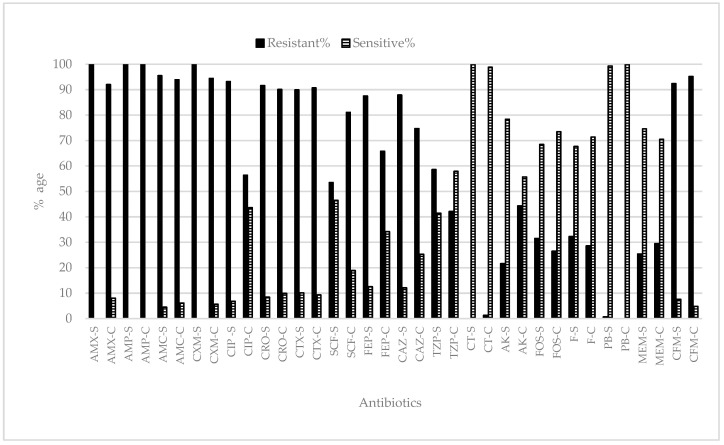
Antibiotic resistance/sensitivity profile of bacterial strains isolated from pediatric patients’ (aged 0–6 years) samples at Shaikh Zayed Hospital and Children Hospital, Lahore, suffering from UTI. Samples from Shaikh Zayed Hospital were represented by suffix “S” and from Children Hospital by suffix “C”.

**Figure 5 antibiotics-12-00395-f005:**
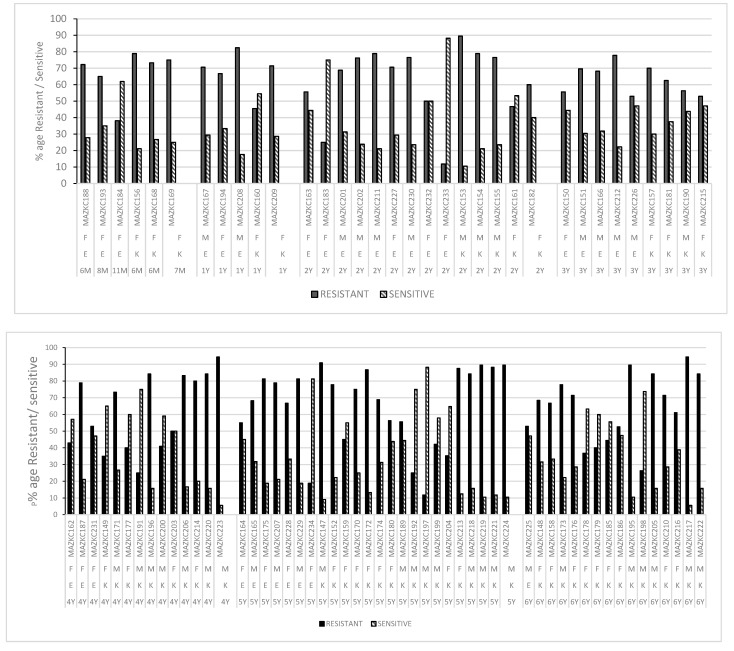
Percentage of antibiotic resistance/sensitivity (against tested antibiotics) in *Escherichia* and *Klebsiella* in relation to age and gender from pediatric patients’ (aged 0–6 years) samples at Children Hospital. E = *Escherichia*; K = *Klebsiella*; F = Female; M = Male; M = Age in Months, Y = Age in years (1 Year, 2 Years, 3 Years, 4 Years, 5 Years, 6 Years).

**Figure 6 antibiotics-12-00395-f006:**
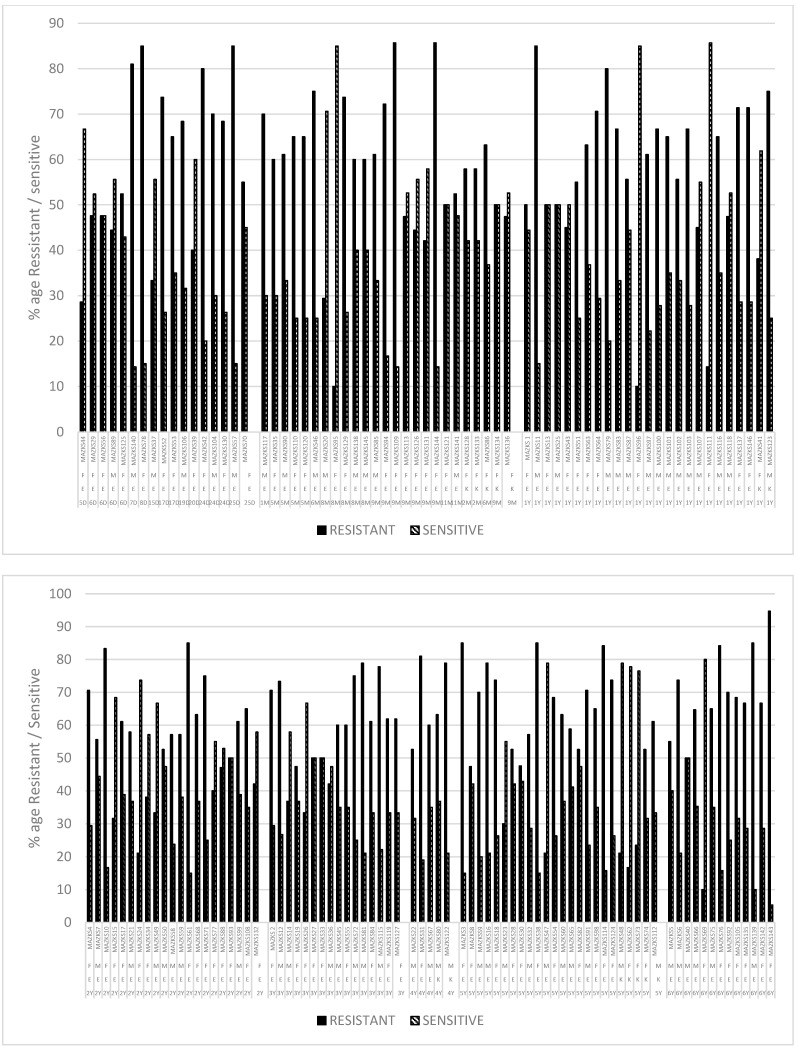
Percentage of antibiotic resistance/sensitivity (against tested antibiotics) in *Escherichia* and *Klebsiella* in relation to age and gender from pediatric patients’ (aged 0–6 years) samples at Shaikh Zayed. E = *Escherichia*; K = *Klebsiella*; F = Female; M = Male; M = Age in Months, Y = Age in years (1 Year, 2 Years, 3 Years, 4 Years, 5 Years, 6 Years).

**Table 1 antibiotics-12-00395-t001:** Total number of samples of pediatric patients (aged 0–6 years) from Shaikh Zayed and Children Hospital, Lahore, Pakistan.

Name of Hospital	Shaikh Zayed Hospital	Children Hospital				
Sampling duration	Samples collected in four months	Samples collected in five months				
Number of samples	15,000	24,750				
Exclusion criteria	Excluded by microscopy < 5 WBC under high power field					
Number of samples excluded	10,800	16,500				
Inclusion criteria	Total number of samples cultured on CLED agar (≥5 WBC high power field)					
Number of samples included	4200	8250				
	Total number of bacterial growth shown on CLED agar					
Number of plates with positive growth	600	750				
Selection criteria	Total number of positive samples for *Escherichia* sp. and *Klebsiella* sp. on CLED agar					
Number of samples selected	146	88				
Gender-wise distribution of samples	Male *n* (%)	Female *n* (%)		Male *n* (%)		Female *n* (%)
	62 (42.46)	84 (57.53)		40 (45.45)		48 (54.54)
Bacterial count	Bacterial count in samples *n* (%)					
>10^4^ CFU/mL	05 (8.06)	16 (19.04)	05 (12.5)		02 (4.16)	
>10^5^ CFU/mL	57(91.93)	68 (80.95)	35 (87.5)		46 (95.83)	

**Table 2 antibiotics-12-00395-t002:** Co-ordinates of the sampling hospitals.

Name of Hospital	Co-Ordinates (DMS)
Shaikh Zayed Hospital	31°30′31″ N, 74°18′31″ E
Children Hospital	31°28′48.36″ N, 74°20′34.8″ E

**Table 3 antibiotics-12-00395-t003:** Antimicrobials used to categorize antibiotic resistance, intermediate resistance, and sensitivity according to CLSI.

Antimicrobial Agents Used with Concentrations			
**Aminoglycosidases** Amikacin (AK, 30 μg Gentamicin (CN, 120 μg) Tobramycin (TOB, 10 μg)	**Quinolones** Ciprofloxacin (CP, 5 μg)	**Polymyxin** Polymyxin B (PB, 300 units) Collistein sulphate (CT, 25 μg)	
**Β-lactam**			
**Penicillins**	**Cephalosporins**		
Co-Amoxiclav (AMC, 20/10 μg) Amoxicillin/clavulanic acid (AUG, 30 μg) Piperacillin/Tazobactam (TZP, 100/10 μg) Amoxicillin (AMX 10 μg)	**Ist generation** Ampicillin (AMP- 10 μg) **2nd generation** Cefuroxime (CXM, 30 μg) **4th generation** Cefepime (FEP, 30 μg)	**3rd generation** Cefixime (CFM, 5 μg) Cefotaxime (CTX, 30 μg) Ceftazidime (CAZ, 30 μg) Ceftriaxone (CRO, 30 μg) Cefoperazone/Sulbactam (SCF, 10/5 μg)	**Carbepenem** (10 μg each) Imipenem (IPM) Meropenem (MEM) Ertapenem (ETP)
Cotrimoxazole ( sulphamethoxazole and trimethoprim) -SXT, 1.25/23 μg	Fosfomycin (FOS, 50/200 μg)	Nitrofurantoin (F, 300 μg)	
Norfloxaxin (NOR, 10 μg)	Pipedemic acid (PIP, 20 μg)	Nalidixic acid (NA, 30 μg)	

## Data Availability

The data presented in this study are available from the authors upon request.

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
