# Peer review of "Antibiotic Drug Resistance Pattern of Uropathogens in Pediatric Patients in Pakistani Population"

_antibiotics, 2023, doi:10.3390/antibiotics12020395_

Round 1
Reviewer 1 Report
In this manuscript, Iqbal et al analyses urine 39,750 samples across two Pakistani hospitals from 0-6 years old patients to evaluate antibiotic drug resistance pattern of urine tract infections (UTI) in pediatric patients. The authors observed the prevalence of both Escheria coli and Klebsiella spp., with higher incidence of one over the other in the two different hospitals. As previously observed, the authors also found higher prevalence of uropathogens in female patients and with its highest occurrence in the month of September. Based on antibiotic resistance profiling across 23 antibiotics, Iqbal et al show that these uropathogens are resistant to most antibiotics, given some exceptions (Polymixin B and colistin sulphate).
Overall, this paper addresses a very important question on the prevalence of antibiotic resistance in pediatric patients with UTIs. Even though the data is very compiling and promising, the authors lack exploring the most of its data. Some improvements can be made in terms of clarity of legends/caption. Furthermore, it would be interesting to discuss even further the possible causes and consequences of high incidence of antibiotic resistance in children.
The following concerns are raised:
Line 45: Kelbsiella to Klebsiella
Line 56: write what MDR, PDR and XDR, WBC stand for
Figure 2: include in the legend the Female samples and which one is for Klebsiella and for Escherichia
Line 111, define what SZH and CHD stand for earlier in the text
Percentages in line 123 and 125 are the same for Motility and citrate utilization. Please verify if that is correct.
Figure 04: Please specify what -S and -C stand for in the caption
Line 174-175 rephrase it.
Figure 05: describe in caption or legend the abbreviations E, K, F, M, 4Y, 5Y, and 6Y
Figure 06: Remove “Chart title” from the 2 panels.
Line 229: The observation that the highest incidence of UTI cases occur in September is interesting. Would be interesting to discuss these results with results from Iqbal et al., (2021).
Line 239: strains were also showed resistance -> strains also showed resistance
The discussion about the increase of resistance against multiple antibiotics overtime shows the possible risks society is facing with the possible raise of super-resistant bacteria. It would be very useful to see the authors generating a plot comparing the data observed for the present study against previous studies with time and %resistance as axes.
Author Response
Dear Reviewer,
We appreciate your efforts in helping us to improve the manuscript. We have incorporated/addressed your suggestions. We are very much satisfied with the editorial process and processing time. Attached is our responses file.
Comments of the reviewers (black text) and our responses (blue text)

Reviewer 2 Report
Authors examined the urine of a total of 39,750 patients from two hospitals, and isolated Escherichia or Klebsiella from 234 patients. Then, isolated bacteria were tested for resistance to 23 antibiotics.
The study design is good, but the presentation of the results needs improvement.
Major comment
・I think it's good that authors specify the hypothesis that "age, locality, and gender have an 53 impact on antibiotic resistance patterns in uropathogens" in Lines 53-54. However, it does not appear that the results are presented in a way that answers this hypothesis. I think it is necessary to use appropriate graphs, such as histograms and boxplots, according to the results authors want to show, instead of just making bar graphs of experimental results.
・Consistency of morphological, biochemical, and molecular (16S rRNA) test results should be demonstrated. Adding extend data summarizing those results (and results of antibiotic resistance test) for each isolated strains would be very valuable.
Minor comment
・The abbreviations (e.g., urinary tract infections: UTI, Shaikh Zayed Hospital: SZH, and Children Hospital: CH), should be used consistently throughout if used.
Line 26-27: I think that "Most of the strains were sensitive (100 26 to ≥90% ) against meropenem, fosfomycin, amikacin, and nitrofurantoin." seems inconsistent with Figure 4.
Line 50-52: I think this sentence is misleading, because it can be read that all uropathogens are resistant to these antibiotics.
Line 55: I don't think authors have verified that the genus Escherichia isolated in this study is E.coli.
Line 69: It's unclear what the "40x" means.
Line 68-69: It is unclear which one is included when WCB = 5.
Line 78-79: I think this sentence has a grammatical problem. It should be rewritten to make sense.
Line 81-82: I think this sentence has a grammatical problem. It should be rewritten to make sense.
Line 84-85: I think this sentence has a grammatical problem. It should be rewritten to make sense.
Line 86-88: It is unclear from what results this conclusion arose.
Line 90-91: I think Figure 1 shows that most patients were found from "urology".
Line 102-103: It is unclear from what results this conclusion arose.
Line 122: It's unclear what the "-ive" means.
Line 124: "triple iron sugar test" might be "triple sugar iron test (TSI test)".
Line 148-149: It's unclear what the "variability" means. And, it is unclear from what results this conclusion arose.
Line 153: "Naladixic acid" might be "Nalidixic acid".
Line 157: It's unclear what the "different antibiotics" means.
Line 167: "Figure04" might be "Figure05".
Line 174-175: I think this sentence has a grammatical problem. It should be rewritten to make sense.
Line 176: It's unclear what the "all of them" means.
Line 183: "Figure05" might be "Figure06"
Line 198: Does the Discussion section start from here?
Line 203-204: I don't think this discussion makes sense unless the authors show the number of males and females in all 39,750 patients.
Line 238-239: I think this sentence has a grammatical problem. It should be rewritten to make sense.
Line 241-242: It's unclear what the "variable resistance pattern" means.
Line 244: [26] might be [16].
Line 281: "at the" is duplicated.
Line 293: "w" might be a typo.
Line 293: If it is a table, it should be treated as a table properly.
Line 319: It would be better if authors describe what results would be considered as Escherichia or Klebsiella.
Line 347: I don't think it's tested to see if the isolated bacteria are ESBL.
Table 1: Please make it easier to see, such as arranging entry in row names.
Table 1: Why is "Total number of samples cultured on CLED agar" less than "Samples collected" minus "Excluded by microscopy ≤5 WBC high power field"?
Figure 1: The vertical axis may be the number of "patients".
Figure 2: Insufficient legend.
Figure 4: Of the 23 antibiotics used in the experiment, only 18 are shown in the graph. Why is this?
Author Response

(The authors gave the same response as above.)
